# The Development of a Care Model for Sarcopenic Obesity in Older Adults: Participatory Action Research

**DOI:** 10.3390/nursrep15100357

**Published:** 2025-10-05

**Authors:** Nuchthida Samaisong, Chomchuen Somprasert, Lisa Pawloski

**Affiliations:** 1Faculty of Nursing, Thammasat University, Pathum Thani 12120, Thailand; nuchthida075@nurse.tu.ac.th (N.S.); lpawloski@ua.edu (L.P.); 2Barefield College of Arts and Sciences, The University of Alabama, Tuscaloosa, AL 35401, USA

**Keywords:** care model, older adults, participatory action research, sarcopenic obesity, transformative behavior

## Abstract

**Background/Problem:** Sarcopenic obesity (SO) is characterized by significant muscle loss combined with obesity, and it is mostly prevalent among older adults. Consequences include a heightened incidence of falls and a greater susceptibility to non-communicable diseases. Thailand currently lacks a care model for SO in older adults. **Objective/Purpose:** This study utilizes participatory-action research (PAR) to develop a care model for sarcopenic obesity in Thailand. **Design and Methodology:** In-depth interviews with 25 older adults with SO and focus group discussions with 12 stakeholders were conducted to develop a preliminary care model. An action research spiral process was utilized with 15 older adults with SO over 16 weeks. **Findings:** We developed a culturally sensitive care model for SO in older adults. This study demonstrates that a participatory-action research (PAR) method for behavior transformation, highlighting health awareness and SO literacy, is crucial for behavior change. **Conclusions and Implications:** The behavior change process using transformative behaviors facilitated internal changes. This approach helps individuals to understand interconnected factors through personal experiences, leading to profound understanding and readiness for deep, continuous, and meaningful behavioral changes.

## 1. Introduction

Sarcopenic obesity is a clinical and functional disorder defined by a concurrent reduction in skeletal muscle mass and function, together with excessive adipose tissue accumulation [1]. This coexisting condition is associated with considerable health risks, particularly in the elderly population, including heightened susceptibility to frailty, cardiovascular disease, osteoporotic fractures, and increased mortality [2].

The prevalence of SO is global phenomenon, including Thailand, where its prevalence is predominantly driven by an aging population and an ongoing obesity epidemic. A meta-analysis of 50 studies globally reported an 11% prevalence of SO in older people aged ≥ 60 years in 2020 [3]. In Thailand, few studies have examined the incidence or prevalence of SO. However, previous research indicates that sarcopenia affects approximately 10% of older adults [4], and 38.4% of the elderly population is obese [5]. Given these high rates of sarcopenia and obesity among Thai seniors, it is plausible to infer the significant prevalence of SO.

Sarcopenia and obesity represent distinct medical conditions; however, they manifest shared pathophysiological characteristics and risk factors, including lifestyle, aging process, the generation of inflammatory cytokines and reactive oxygen species, and endocrine changes. Furthermore, these two conditions collaborate synergistically, mutually amplifying each other and giving rise to a deleterious cycle of exacerbation [6].

The management and treatment for older adults with SO present unique challenges, as approaches differ between those used for individuals with either obesity or sarcopenia alone. Nutritional guidance must balance energy intake while simultaneously increasing protein consumption. Additionally, exercise recommendations emphasize both increasing overall energy expenditure and enhancing muscle strength [7]. Due to the vague symptoms and gradual decline associated with SO, there is no established screening standard in clinical practice. This lack of standard has led to underdiagnosis and insufficient prevention, treatment, and standardized care for SO. Furthermore, encouraging health-behavior changes in elderly individuals requires a focus on participation, attitudes, and a motivation to enhance well-being [8]. Caring for elderly individuals with SO requires a specialized approach that considers both the specific characteristics of the disease and the unique attributes of the affected population. Effective and sustainable behavior modification requires the patient’s awareness of their condition and motivation for self-directed change. This process aligns with transformative behavior modification, which has not previously been examined in obese older adults with muscle deficiency. Therefore, it is essential to develop a care model for elderly individuals with SO to ensure disease-specific and patient-centered care, ultimately optimizing the effectiveness of treatment and management.

Transformative learning theory, originally proposed by Mezirow in 1975 [9], explains how major life events change a person’s perspective and understanding of the world. According to Mezirow’s 1991 refinement of the theory, learning occurs when individuals face problem-solving situations that are filtered by their worldview. When individuals critically reflect on the assumptions and expectations underlying their life experiences, they may recognize that these assumptions are flawed or too narrow. This could lead to a revision of their perspective, which may result in behavior change. Essentially, a shift in internal interpretation drives a change in external actions [9].

Transformative learning has been applied to health-behavior modification by enabling participants to engage in the co-construction of knowledge in collaboration with facilitators, thereby integrating new information and expanding existing meaning schemes. The process typically begins with patients receiving information regarding their conditions, underlying causes, treatment options, and the influence of specific behaviors. Subsequent discussion and dialog stimulate critical reflection, allowing for learners to evaluate the credibility of differing sources of information. Through this process, participants can articulate how recommended behaviors align with their daily lives and share insights with others about potential challenges and solutions. As a result, participants improve their compliance, self-efficacy, and sense of empowerment [10].

Participatory-action research (PAR) is a research method that encourages the involvement of stakeholders to enhance research outcomes and foster positive changes in the lives of participants and their communities [11]. In this study, a care model for SO in older adults prioritizes patient-centered care and involves all stakeholders to ensure a more targeted and effective solution to the problem. A key characteristic of PAR is its emphasis on capacity building and empowerment through collaborative knowledge generation. By engaging participants as co-researchers in selecting appropriate data-gathering techniques, PAR enhances problem-solving and addresses relevant issues. Involving older adults in PAR offers a promising approach to understanding and addressing complex health and social challenges while simultaneously strengthening individual and community capacity [12].


**Research Questions/Objectives**


This research aimed to design a creation process and develop a care model for older adults with SO through a PAR spiral.

## 2. Materials and Methods

### 2.1. Design

This study consisted of 2 phases. In Phase 1, a qualitative descriptive design was employed, utilizing in-depth interviews and focus group discussions to gather preliminary data for activity development. In Phase 2, participatory-action research (PAR) was adopted as the methodological approach to develop the care model. Data collections were conducted from March to August 2023.

### 2.2. Context/Setting

The study area selected included a rural Muslim community in Chumphon Sub-district, Ongkharak District, Nakhon Nayok Province, due to prevalent obesity and a significant elderly population. The community has 738 older adults (16.70% of the population), with 50% being overweight or obese [13]. Most of the population primarily engaged in agriculture, particularly rice farming. Both men and women work together in fields performing manual labor, although men typically performed more physically demanding tasks. Most individuals ceased agricultural work as they grew older and remained at home, where physical activities mostly consisted of light housework.

### 2.3. Sampling Procedures

Participants who were older adults with SO were accessed using purposive sampling and snowball sampling methods. All stakeholders were selected by purposive sampling.

Inclusion and Exclusion Criteria

Phase 1: The participants were separated into primary and secondary target groups. The primary target population was older adults with SO who met the following inclusion criteria: (1) age of 60 years or older; (2) diagnosis with SO based on the Asia Working Group of Sarcopenia 2019 (AWGS) sarcopenia diagnostic criteria and the World Health Organization (WHO) obesity criteria on body mass index (BMI) for Asian populations; and (3) Thai Mental State Examination (TMSE) score > 23. The following exclusion criteria were applied: (1) CKD stage 3 or higher, defined as a GFR of 60 mL/min/1.73 m^2^ or lower; (2) undergoing cancer treatment; (3) life-threatening comorbidities, including patients with unstable ischemic heart disease or congestive heart failure; (4) muscle problems, such as myasthenia gravis; (5) hearing, vision, or communication problems; and (6) known psychiatric problems.

The secondary target groups were all stakeholders involved in the SO problem, who were supporting the transformative behaviors of older adults with SO, including the following: (1) family members of older adults with SO; (2) a registered nurse from the Chumphon Subdistrict Health Promoting Hospital; (3) public health volunteers; and (4) a community leader.

Phase 2: The participants in Phase 2 met the same inclusion and exclusion criteria as the primary target population in Phase 1, with additional considerations related to Barthel Activities of Daily Living (ADL) score > 12. Some participants in Phase 2 were the same individuals who participated in Phase 1 of this study.

### 2.4. Participant Selection

In Phase 1, the researcher analyzed medical records from Chumphon Subdistrict Health Promoting Hospital to identify the target population, which focused on elderly individuals. Eligible participants were screened during their appointments at the hospital’s chronic disease clinic. Contact information was collected for follow-up in-depth interviews. Additionally, the researcher collaborated with village health volunteers to locate suitable participants. A snowball sampling approach was employed, wherein interviewed individuals recommended others aged 60 and above with an obesity profile for further screening. Interviews were conducted with participants at locations convenient for them. Eligible participants who met the inclusion criteria were asked to fill out an informed consent before proceeding. During the interview, which lasted approximately 60 min, participants gave consent for an audio recording alongside written notes to ensure accuracy and facilitate data analysis.

The secondary target groups were selected from all stakeholders who voluntarily agreed to participate in the study. The purpose of the activities was to corroborate data from in-depth interviews and to gain additional insights from all stakeholders.

In Phase 2, participants were selected from Phase 1, and eligible participants were screened during their appointments at the hospital’s chronic disease clinic.

### 2.5. Interview Questions/Tools

Screening and eligibility assessments included measurements of body composition (body fat and muscle mass) and body weight using bioelectrical impedance analysis (BIA). Height and weight data were used to calculate BMI. Handgrip strength was measured using a dynamometer. Functional ability in performing daily activities was evaluated using the ADL scale. Cognitive function was assessed using the TMSE, a culturally adapted and enhanced tool for evaluating cognitive status in the Thai population, with demonstrated reliability (Cronbach’s α = 0.77) [14]. Participants also completed sociodemographic questionnaires. Data collection involved both in-depth interviews and focus group discussions, guided by protocols validated by a panel of five experts (3 nursing professors who are experts in qualitative research and geriatric nursing, 1 physician specializing in qualitative research and family medicine, and 1 expert in health-behavior modification). Prior to data collection, the researcher reviewed relevant documents and health records of participants. During the in-depth interviews, focus group discussions, and group activities, field notes were taken and observational techniques were employed to complement the data. Individual interviews lasted approximately 60 min, while each focus group session lasted about 120 min.

### 2.6. Data Analysis

In Phase 1, this study used in-depth interviews among older adults with SO to explore their perceived problems, related factors, care needs, and other issues pertinent to creating healthier older adults with SO in the community. The interview questions were open-ended, with 21 probing questions. The guidelines for the in-depth interview questions were based on the behavioral risk factors for SO and non-pharmacological strategies for managing SO in older adults. The researcher conducted focus group interviews involving all community stakeholders to triangulate the data, parallelling content analysis with data collection until data saturation was achieved. In the process of conducting in-depth-interviews and focus group interviews, the researcher took field notes and used the technique of observation during data collection. All information obtained from the in-depth interviews and focus group interviews was used to develop the preliminary care model for SO in older adults in the PAR process.

In Phase 2, the researcher followed the PAR process in the care model development. This process consists of four steps: planning, acting, observing, and reflecting [15]. Re-planning began after the reflecting stage, initiating a new cycle of the spiral process that would continue until the care model attained suitability. Suitability was demonstrated by its practical applicability and by showing that it benefited participants by promoting healthy sarcopenic obesity management. The researcher conducted content analysis in parallel with the PAR process to analyze the development of care models. Participants who met the inclusion/exclusion criteria and provided informed consent were appointed to the meeting room at the subdistrict hospital for research activity. The researcher served as the facilitator of the research process, ensuring that all participants were actively involved in developing the care model.

### 2.7. Data Rigor

Lincoln and Guba (1994) [16] based trustworthiness on four criteria: credibility, dependability, transferability, and confirmability. ***Credibility*** was maintained by triangulation and regular participation in various community activities to eliminate bias and understand the important issues of the participants. ***Data triangulation*** was maintained by obtaining data from different groups, including older adults with sarcopenic obesity, family members, registered nurses, public health volunteers, and community leaders. Additionally, the activities in the PAR process were divided into two groups as a different technique of data triangulation. ***Investigator triangulation*** was maintained by the research advisors, expert researchers involved in the study, by consulting and reflecting on the data throughout the process of data collection. ***Methodological triangulation*** was maintained by using multiple data sources, including in-depth interviews, focus group interviews, field notes, observation, and documentation in this study. ***Dependability*** was maintained by checking and confirming the research process to verify that procedures were followed correctly with advisors. ***Confirmability*** was maintained by conducting a check of the peer debriefing after activities at every stage of the research, taking note of every detail and asking a qualitative expert advisor to share details, critique, and comments on the information/findings to determine whether the information received was accurate and relevant to the research question and, if so, how. ***Transferability*** was maintained by explaining the findings clearly with comprehensive descriptions based on the information drawn from the perspectives and experiences of the participants, research methods, interpretations of the study results, and peer debriefing.

### 2.8. Ethical Considerations

This study was conducted in accordance with established ethical guidelines for research. This study was conducted in accordance with the Declaration of Helsinki, and was approved by the Human Research Ethics Committee at Thammasat University (Science) under certificate No. COA 015/2566 on 2 March 2023. Following approval from the ethics committees, the researcher met with participants to explain the study in detail. Prior to obtaining written informed consent, participants were provided with comprehensive information regarding the research protocol, objectives, potential benefits, risks, and measures to ensure confidentiality. They were clearly informed that participation was entirely voluntary and that they could withdraw from the study at any time without penalty. This information was communicated through both verbal explanations and written information sheets.

## 3. Findings

The findings were extracted from in-depth interviews with 25 older adults with SO in the community and the focus group interviews with 12 stakeholders. The participants were aged 62–88 years, with a mean age of 69. Most of the participants were Muslim (93.3%), female (88.8%), and had completed primary education (86.7%).

The content analysis results revealed five categories: (1) Lack of sarcopenic obesity perception and awareness. (2) A deficiency in wellness literacy. (3) Poor eating and exercise behaviors. (4) Lack of family support for healthy behaviors. (5) Desire to be independent, without external assistance [17] (Table 1).

The participants in Phase 2 consisted of 15 older adults with SO who met the defined criteria. This study divided the participants into two groups, Group 1 (seven participants) and Group 2 (eight participants), to enhance the effectiveness of the group process and data triangulation. Additionally, insights from the implementation in Group 1 were used to refine the approach in Group 2, ensuring a comprehensive care model. Most of the participants in Phase 2 were females (86.7%). The mean age of the participants was 67.9 years. The lowest age was 62 years and the highest age was 78 years; 80% of the participants were Muslim. Most of the participants completed primary education at 86.67%. All participants had chronic diseases, including dyslipidemia (100%), hypertension (87%), diabetes mellitus (53%), heart disease (7%), and other chronic diseases (7%). All participants remained in the study until its completion.

The development of the care model for SO in older adults required over four cycles, and each cycle of the activities took 4 weeks. The activities were held 1 day per week, and the duration of each activity was approximately 90–120 min. The activities over the 16-week period are display in Figure 1.

Cycle 1 included sarcopenic obesity literacy facilitation, raising awareness of SO including understanding risk factors, health impacts, and solutions for managing problems. For Cycle 1, the researcher reflected on the participants’ responses, linking theories to evidence-based outcomes on themselves or fellow participants. The researcher also observed and implemented two activities related to food and exercise. Anthropometric measurements and muscle strength risk factors were taken and shared with participants. This created awareness about personal health status and means for finding solutions together. The researcher and the participants set short-term goals together over the 4 weeks of Cycle 1 and long-term goals throughout the 16 weeks of participation in the program.

Cycle 1 results show that the participants gained an understanding of their body composition and the issues related to SO. They also learned strategies for managing SO. It was evident that the participants possessed knowledge about SO and were aware of its health impacts. The data obtained from the reflections led to action plans in Cycle 2, including how to implement that knowledge into practice in daily life. For example, one participant stated, “I understand the doctor’s information on a high-protein diet and daily exercise but remain unsure which meals are high in protein.” (G202).

Cycle 2 involved the enrichment of healthy behavior and the implementation of the knowledge gained from Cycle 1. For Cycle 2, the researcher inquired about the needs, activity goals, and activity patterns that would help manage and prevent complications from SO. During this cycle, the researcher also observed and implemented five activities including muscle strengthening, understanding hidden calories in high-sugar beverages, exercises on reducing calories in meals, a cooking contest, and involvement with family support. As with Cycle 1, the researcher again took anthropometric measurements and muscle strength indicators and shared the results with each participant for their individual consideration.

Cycle 2 findings indicate that participants could apply their knowledge to practical situations in daily life. They observed healthy meals prepared by group members, created their own exercise equipment, and gained an understanding of the calorie content of foods they regularly consume. Participants with SO modified their lifestyles, showing a good attitude to practice health behaviors. The data derived from Cycle 2 informed improvements and action plans for Cycle 3, which focused on participants maintaining healthy behaviors.

Cycle 3 focused on the importance of independent daily activities and applying the knowledge gained as a regular part of their lifestyles (cultivated behavior modification). For Cycle 3, the researcher taught the participants to recognize the value of performing daily routines independently through six activities throughout the four-week cycle. These included 1, “Freedom in Life”, for which the objective was to allow for participants to see that “The freedom of human beings to perform daily activities on their own is the basis of all happiness in life.” Activity 2 was “Make A Choice” in order “to assist with good diet choices for managing SO. Activity 3 was “Recognize Knee Pain”. Activity 4 was “Miracle of Weight” to encourage weight-bearing exercises to strengthen muscles. Activity 5 was “Thai Hamstring Stretching Device” to increase the efficiency of movement and prevent falls. As with Cycles 1 and 2, the researcher again collected anthropometric measurements and muscle strength indicators and shared the assessment results to each participant for their consideration and reflection. For Cycle 3, the participants reflected that they were more aware of the importance of being able to perform daily tasks independently. They also recognized the significance of managing their diet and engaging in exercise to maintain muscle strength and lose weight. Again, the data obtained from the Cycle 3 reflections of the older adults led to improvements and action plans in Cycle 4. In this next step, the participants needed to promote knowledge sharing in the community to create sustainable behavioral change. One participant said, “My relative is obese and inactive, so I advised him to exercise and control his diet. I will follow up next week..” (G104).

In Cycle 4, behavior changes and knowledge from participation in the project promoted knowledge sharing in the community to create sustainable behavioral change. Social interactions among the older adults provided them with the opportunity to exchange knowledge and experiences related to behavioral modifications aimed at rehabilitating those with SO. The goal was for the participants to become role models who demonstrated enhanced physical capabilities and strength. For Cycle 4, two activities were implemented and included: (1) “Traffic Lights” and (2) “Good Memories, Good Emotions, and Happy Life”. For “Traffic Lights”, participants were asked to summarize knowledge about appropriate foods for older adults with SO. The second activity used the “Stroop Test”. This test asks patients to read and remember the colors they saw. In addition, the participants were invited to disseminate knowledge and guidelines to other seniors in the community. In this way, they acted as role models to the community. Again, anthropometric and muscle strength indicators were taken and shared with participants. The output from activities in Cycle 4 ultimately led to better reported health among participants. According to one participant: “I think I have to take care of myself when it comes to eating. The younger ones can’t eat like me. I’m old. They’re still young. Now, I have to make my choices. Eating less of whatever is bad. Before, when my daughter made bale fruit syrup, I’d eat ‘till my sugar was high. I know to eat only a little now when she makes it. I have to take care of myself. It’s not possible to just have anyone tell me and work everything out as I want.” (G201).

The development of healthy sarcopenic obesity among older adults was based on a participatory research spiral, which is designed to explain the mechanisms of change in the continuous and dynamic development of older adults until the desired goals are achieved. The behavioral transformation cycle for older adults comprises four steps: Step 1: Facilitating sarcopenic obesity literacy. Step 2: Enriching healthy behaviors. Step 3: Cultivating behavior modification. Step 4: Sustaining dynamic behavior change.

The output of the care model for older adults with SO was “healthy sarcopenic obesity”, which is a condition where, despite having SO, older adults have no physical, emotional, and psychosocial effects from the disease, but remain able to perform activities in daily living. In addition, they are able to live freely in the community with appropriate healthcare behaviors and an accurate personal perception of health. In addition, by gaining literacy about sarcopenic obesity and general health awareness, the participants were empowered to modify their behaviors, effectively manage their condition, prevent complications, and serve as a role model for others. Figure 2 displays the entire care model developed with the inputs, four-cycle process, and output of healthy sarcopenic obesity.

## 4. Discussion

In developing a care model for older adults with SO, the researcher utilized baseline data from participants, including perceptions of SO, health effects, care needs, and required resources. These insights were gathered from older adults with SO. This study employs the PAR methodology, integrating the healthcare development process. Participants actively engaged in each cycle, contributing to planning, implementation, observation, and reflection. The researcher facilitated the study through four iterative cycles.

The care model development was consistent with the development of other care models. For example, the model was developed similarly to the service provision model for older adults in Buriram, Thailand [18]. The risk factors contributing to SO in older adults within this community do not differ from those described in theoretical models, which include high energy intake and low protein consumption [19]. However, this study provides deeper insights into attitudes and reasons for consuming high-energy, low-protein diets. It was found that older adults tend to consume large quantities of rice, fruit, and sweets while lacking awareness of the importance of protein intake, despite the abundant availability of protein-rich food sources in the community. These findings can inform the development of tailored behavioral interventions to promote appropriate dietary habits among older adults in this community.

The researcher used PAR guidelines in developing the care model for older adults while emphasizing stakeholder participation at every step, which helped to create sustainable solutions for problems. PAR is also frequently used to develop care models for older adults in many dimensions in Thailand, such as promoting independent living [20], holistic health promotion, and hypertension patient care [21]. PAR is an approach aimed at driving context-specific change or development through shared decision-making and the promotion of equality among participants. It involves groups in action, reflection, and iterative change. PAR has been shown to be effective in enhancing patient-centered care and promoting sustainable behavioral change [22].

The activities employed health-behavior modification processes along with transformative learning guidelines. The transformative learning process created sustainable behavioral change [23]. Health-behavior modifications created internal personal change and helped the participants to understand the interconnected causes and factors. These changes helped to prepare participants to listen and make in-depth changes that were sustainable. Healthcare providers may employ transformation approaches in the management of other chronic diseases common among older adults and support the modification of self-care behaviors to control symptoms or prevent complications, such as diabetes, hypertension, and hyperlipidemia.

The first step of transformative behavior was the sarcopenic obesity literacy facilitation process, which focused on producing awareness of health conditions, especially SO. According to research reports, health-behavior modification in older adults should begin by creating perception and awareness about disease and health, including potential effects, which will lead to searches for information as a guideline for disease treatment with correct and appropriate health-behavior transformation [24].

The next step was the enrichment of healthy behavior. Older adults face challenges in behavioral change due to the long-standing nature of their lifestyle habits. Additionally, they may experience limitations in learning, hearing, vision, and physical abilities, all of which can impact behavioral transformation. Therefore, understanding the nature of behavioral change in older adults is a crucial starting point. Efforts to modify behaviors and achieve desired outcomes should prioritize motivation, as it has been shown to be more effective than education, awareness-building, or health education alone [25]. Older adults experience physical, emotional, and social changes that affect personal health transformation. The activities organized in this study were the result of joint planning by the researchers and participants concerning learning facilitators and barriers in older adults, which resulted in cooperation with the activities and positive attitude to behavior change. The data from this study are consistent with data from a meta-analysis of positive attitude toward health behavior modification among older adults, which created sustainable modification [26].

The third step involved cultivating behavioral transformation. This process integrates behavioral change into daily routines and requires motivation for sustainable transformation. In this study, the researcher emphasizes the importance of independence in daily activities as a key factor contributing to quality of life and overall well-being. This independence, in turn, relies on adequate muscular strength. The key considerations in this step align with Khodabakhsh’s study, which found that quality of life and life satisfaction in older adults correlate with physical health status and the ability to independently perform activities of daily living [27].

The group activities in the older adults were a major success factor in behavioral transformation. The participants were older adults with similar health problems, social contexts, and lifestyles. Therefore, the participants had similar interests in health conditions and behavioral modification. Participation in group activities helped the older adults exchange knowledge and experience, facilitating positive behavioral transformation. The data are consistent with the findings, which reveal that group activities or social support improved health-behavior modification among older adults [28].

The last step was summarizing knowledge from participation and promoting knowledge sharing in the community to create sustainable behavioral change. This process promoted the use of analytical thinking techniques, including knowledge and experience sharing. This was consistent with Panich’s learning technique, which states that learning from exchanges, conversations, practice, and teaching others creates memories and facilitates learning in 90% of participants [29]. The output derived from this process is that participants have analytical thinking techniques. This process is commonly referred to as “dynamic behavior change”.

The output of this care model was “Healthy sarcopenic obesity”, which is consistent with the desired characteristics leading to physical, psychological, social, and spiritual happiness among older adults, or healthy aging. The WHO defines “healthy aging” as “the process of developing and maintaining the functional ability that enables well-being in older age.” The ability to perform duties is related to the ability to meet basic needs in daily life, the ability to learn, develop, and make decisions, and the ability to perform activities independently, build and maintain relationships with others, and aid society [30].

Recommendations for implementation

Health-behavior transformation among older adults, facilitated through a transformative learning process, may lead to the development of positive attitudes toward behavior modification.

Limitations in implementation

Implementing this care model or scaling-up its findings should consider the relevant community data, and the healthcare model should be adapted to suit the specific context of each community. The long-term sustainability of behavior changes was not assessed.

## 5. Conclusions

This research uses participatory-action research to develop a care model for older adults with sarcopenic obesity. This research uses baseline data from the community and applies theoretical data based on the context and needs of community members. This processes successfully increased health and sarcopenic obesity literacy among participants, leading to health-behavior transformation. Ultimately, this increased faith in behavior modification and sustainable health habits, resulting in improved health and overall well-being for older adults with sarcopenic obesity.

## Figures and Tables

**Figure 1 nursrep-15-00357-f001:**
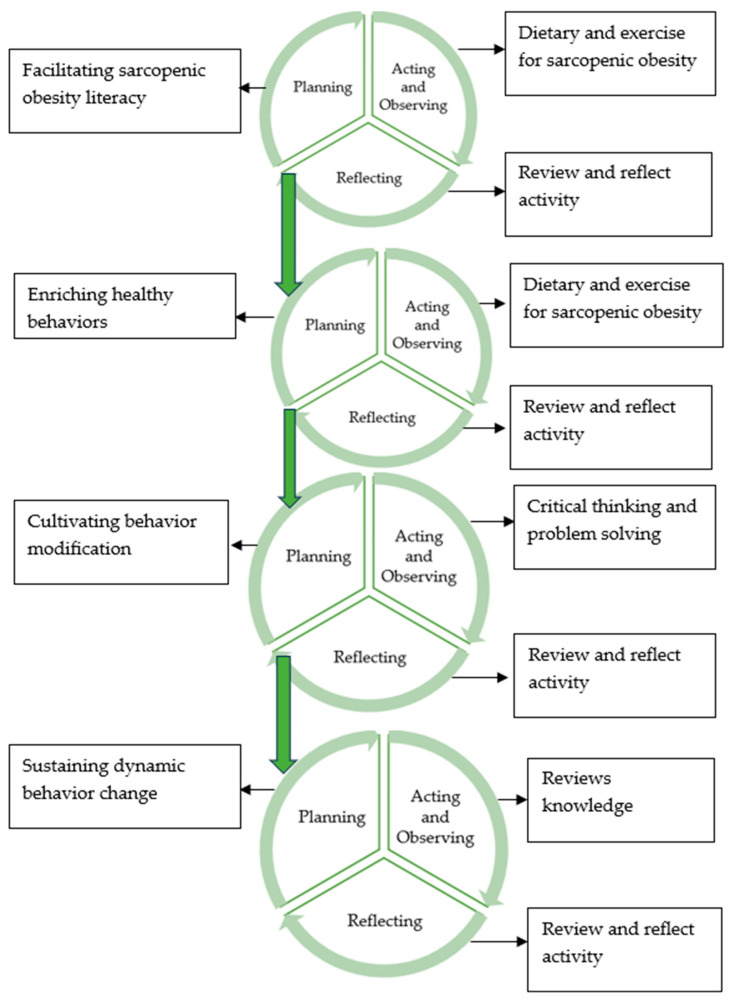
Participatory-action research process for developing a sarcopenic obesity in older adults care model.

**Figure 2 nursrep-15-00357-f002:**
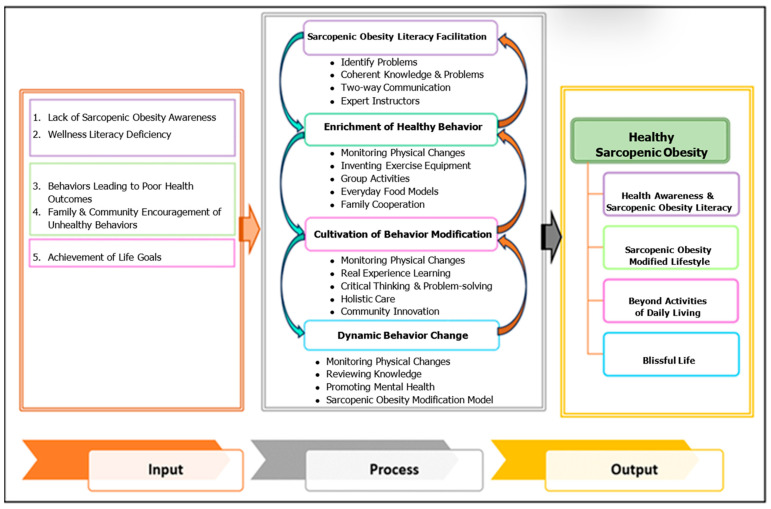
Care model for sarcopenic obesity in older adults.

**Table 1 nursrep-15-00357-t001:** Sample quotations from participants in each category [17].

Categories	Sample Quotes
Lack of sarcopenic obesity perception and awareness	“I am familiar with obesity and fully aware that I am currently obese. However, I have never encountered the terms ‘sarcopenic obesity’ or ‘sarcopenia’ before. What do they mean?” (A08)
Deficiency in wellness literacy	“I’ve never thought about losing weight. When the doctor told me to lose it, I didn’t want to. This weight is not a problem. My legs and knees are normal for an older person. It’s not related to being fat or thin.” (A24)
Poor eating and exercise behaviors	“Most of my meals are soupy. I don’t eat much meat. I don’t like fish, any meat, eggs and milk. I can eat just a bit. I don’t eat vegetables often. I mostly eat soup with rice. I like fruit, I eat three kilograms of ripe mangoes over two or three days.” (A06)
Lack of family support for healthy behaviors	“My sons/daughters never forbid me to eat anything. They only remind me to eat. They buy me delicious things. They know I have diabetes, but they say just to eat, I won’t die. They don’t forbid me, they’re afraid I’d starve. If I’m sick, they take me to the doctor and say to eat because you can’t eat when you’re dead.” (A16)
Desire to be independent	“I’ve dreamed in my sleep before that I could walk and didn’t need a walker. It was so nice in my dream. I was happy. I want it to be real.” (A13) “I want to be able to walk and do normal activities. I want my body to be strong. I don’t want to be a burden to my sons or daughters.” (A20)

## Data Availability

For further details regarding this research, please refer to the following link: https://digital.library.tu.ac.th/tu_dc/frontend/Info/item/dc:313545, access on 1 June 2023.

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
