# Peer review of "The Development of a Care Model for Sarcopenic Obesity in Older Adults: Participatory Action Research"

_nursrep, 2025, doi:10.3390/nursrep15100357_

Round 1
Reviewer 1 Report
Comments and Suggestions for Authors
Dear authors,
This reviewer is not a native English speaker, but I am able to notice several inconsistencies in the writings that makes difficult to review this manuscript. The need for a national framework for Development of a Care Model for Sarcopenic Obesity in Older Adults is so important, that I recommend the editor to publish this paper, but without the inconsistencies that I, as well as, a reader interested in the topic.
Metholodical questions
Some readers would like to test whether they can do in their country.
Phase 1 criteria:
- How much is the BMI cutoff for Asians who are 60 years old to be considered obese?
- AWGS and WHO acronyms, as well as any other acronym should be defined in the text of the manuscript. There is no need to define BMI or TMSE under the "Interview Questions/Tools" heading: they were earlier defined!
- Why are persons with overweight excluded?
- Which muscular and pschiatrics conditions were excluded? who did state such conditions?
- Why persons under cancer treatment were excluded?
- What does "moderate to severe" mean in the context of kidney function?
- Which "life-threatening" conditions are needed to be excluded?
- Why persons with sight and hearing excluded?
- Were persons unable to communicate included?
- Who do the researchers know that there are 738 older aldults living in Chumphon?
- Please draw a flowchart starting with the population (N=738 older adults), up to the effective population participating in Phase 1 and Phase 2: N with reasons for not participation, loss in follow-up, exclusions and deaths.
Participants selection
- Who did analyse the medical records? Could you bring her initials? What did she analyse?
- When the authors say "Subdistrict Health Promoting Hospital", do they really mean is "Chumphon Health Promoting Hospital"?
- Are there any patients not attended in the Hospital? If so, who are them?
Interview Questions/Tools
- Who are the persons involved in the "expert panel"? What are their expertise on? Do they have any conflicts of interest?
Ethical considerations
- Please state once again the IRB and approval certificate under this heading.
Findings
- There is no description of the sex nor gender of the participants under study.
- Sarcopenia: In line 212, the authors write: "Most of the participants ... had completed primary education (86.7%)". However, the author show as an example the following quote: "However, I have never encountered the terms
perception and awareness 'sarcopenic obesity' or 'sarcopenia' before. What do they mean?". Does the subject is really aware on the "sarcopenia" concept because he/she does not know the meaning of the word? If he/she does not know the word, perhaps he/she knows the "concept" for example: lack of muscle mass? or lack of strength? or something like that. Furthermore, I don´t expect someone with university education who studied, social scients, laws or engineering be familiar with sarcopenia. Some languages use prefixes, sufixes and stems that makes easy complex concepts. This reviewer is a native Spanish speaker, and both Swedish and English foreign speaker: for sarcopenia, there are no similar vocabulary structures for any of these three languages. Are there any similar vocabulary structures in Thai language for sarcopenia? Individual A20, is aware he/she is weak? - Individual A13 is in a wheelchair. Why is he/she sitting there?
- Can the authors produce any examples of education materials in their original languague as wells as its English translation?
Findings
- I see that there are too many descriptions of methods in this section: for example, is a novel detailed description of the four cycles. Why are they described under Findings section and as a section under Material and methods one? The reader would like to know more about the output of the process?
- Can the authors produce hard outcomes and/or functional outcomes? Were changes in AWGS, BMI, diet and/ TMSE could be recorded to produce tables? If not, could you explain under a Limitation heading?
- I don't understand the meaning of this sentence: "The prevalence of SO is global phenomenon including in Thailand, where the prevalence is driven primarily by the aging population and the obesity epidemic." Does the sentence brings information to the reader?
- Regarding this sentence: "A meta-analysis of 50 studies globally reported an 11% prevalence of SO in older people in 2020." I ask how old were the people under study in the meta-analysis?
- Regarding this sentence: "In Thailand, there is a lack of studies...". This reviewer does not recommend the authors to say there is a "lack" of literature. The "lack of literature" really means that the authors tried to look for literature and they failed! A very interested reader could find literature that the authors could not! Furthermore, the next sentence contradicts this statement: there is literature regarding this issue, but it is not written in English. It seems that the authors have found literature.
- Regarding this sentence: "Sarcopenia and obesity represent discrete medical conditions"; what do the authors mean with the word "discrete"? Both conditions can be deathly, particularly in advanced stages! Please use another appropriate word!
- "Managing and treatment for older adults with SO presents ...", as far I understand, mananaging and treatment, aren't the same thing? if they are not, they mean "plural" and the right verbal form of the verb present should also be plural, ergo, "present" and not "presents".
- Regarding these two sentences: "Nutritional guidance must be tailored to balance energy intake control while simultaneously increasing protein consumption. Additionally, exercise recommendations emphasize both increasing overall energy expenditure and enhancing muscle strength." As far I understand, the second sentence does not bring additional information: the treatment requires to increase in protein consumption, energy expenditure and physical activity. Increased physical activity goes together with increased energy expenditure.
- Be consistent with the number of decimals! The authors can use the number of decimals they wish, but this reviewer recommends one decimal.
Author Response
All response in attached files

Reviewer 2 Report
Comments and Suggestions for Authors
Dear Authors,
Thank you for the opportunity to review your manuscript on developing a care model for sarcopenic obesity in older adults using participatory action research. This study addresses a significant healthcare gap and employs an appropriate methodology for developing a care model. Below is my detailed evaluation, along with suggestions for improvement.
Title and Abstract
The title accurately reflects the study content. The abstract effectively summarizes the PAR approach and key findings.
Suggestions for improvement:
- Consider adding the specific setting (rural community) to provide context for readers
- Include participant numbers (n=25 in Phase 1, n=15 in Phase 2) to clarify the scope
Introduction
The introduction establishes clinical context well and appropriately justifies the need for culturally sensitive care models.
Suggestions for improvement:
- Lines 31-37: If specific SO prevalence data for Thailand are available, include them; if not, the inference approach is reasonable
- Consider briefly mentioning why PAR was chosen over other care model development approaches
Materials and Methods
The PAR methodology is well-suited for this type of care model development, and the two-phase approach is logical.
Suggestions for improvement:
- Lines 140-153: Expand description of what constituted "suitability" criteria for the care model
- Consider including a flow diagram showing participant progression through phases
- Lines 177-198: The data rigor section is comprehensive - well done
Results
The presentation of qualitative findings through the five categories is clear and well-supported by participant quotes.
Suggestions for improvement:
- Table 1: Excellent use of authentic participant voices
- Consider briefly reporting participant retention across the 4 cycles
- The cycle-by-cycle description provides good insight into the iterative process
Discussion
The discussion appropriately contextualizes findings within existing literature and theory.
Suggestions for improvement:
- Lines 341-356: The connection to transformative learning theory strengthens the theoretical foundation
- Consider adding a brief statement about which model elements might be most/least transferable to other contexts
- The limitations section could mention that the long-term sustainability of behavior changes was not assessed
- Consider discussing how healthcare providers might implement similar participatory approaches
- The "healthy sarcopenic obesity" outcome concept is innovative and could be elaborated
Overall Assessment
This manuscript demonstrates a thoughtful application of PAR methodology to develop a culturally appropriate care model for an understudied population. The findings provide valuable insights for healthcare providers working with older adults with sarcopenic obesity, particularly in similar community settings. The methodology is sound, the analysis is thorough, and the results contribute meaningfully to the literature on community-based care model development.
Sincerely,
Reviewer
Author Response
Response as attached file

Reviewer 3 Report
Comments and Suggestions for Authors
Thank you for the opportunity to review this manuscript. I offer the following feedback to strengthen the research:
- Abstract: this section indicates that 15 older adults participated in the spiral process, while the Findings section reports 25 older adults who participated in in-depth interviews and 12 stakeholders who participated in focus group interviews. These discrepant sample sizes require clarification and verification.
- Research concepts and definitions
Given that Participatory Action Research (PAR) serves as the research methodology, the authors should present a detailed methodological framework that includes PAR's fundamental principles, procedural steps, and specific justification for its appropriateness in the SO context.
- Research questions/ objectives
Given the substantial body of existing research on similar topics, a comprehensive literature review should be conducted prior to establishing the research objectives. The authors need to clearly identify the research gap this study fills and articulate the distinctive strengths or novel contributions of their work.
- Sampling Procedures
Several methodological details require clarification: (1) the specific sampling timeframe, (2) the timing of when interviews were conducted, and (3) comprehensive sociodemographic characteristics of participants in Phase 1 and Phase 2.
- Findings
The study included 12 stakeholders (e.g., family members, nurses, health volunteers, and community leaders) in the research process. The manuscript should provide explicit description of how these stakeholders contributed to and facilitated the development of the care model for SO in older adults across cycles 1 through 4.
- Several sections of this manuscript (including objectives, keywords, and conclusion) appear to overlap with a doctoral dissertation published in 2023 at THAMMASAT University, titled "THE DEVELOPMENT OF A FAMILY AND COMMUNITY BASED CARE MODEL FOR SARCOPENIC OBESITY IN OLDER ADULTS". This potential overlap requires clarification.
Author Response
Response as attached file

Round 2
Reviewer 3 Report
Comments and Suggestions for Authors
All my concerns from the previous review have been addressed. The authors have made appropriate revisions that have improved the manuscript's quality.